# Predicting Intention to Receive COVID-19 Vaccination in People Living with HIV using an Integrated Behavior Model

**DOI:** 10.3390/vaccines11020296

**Published:** 2023-01-29

**Authors:** Bramantya Wicaksana, Evy Yunihastuti, Hamzah Shatri, Dicky C. Pelupessy, Sukamto Koesnoe, Samsuridjal Djauzi, Haridana Indah Setiawati Mahdi, Dyah Agustina Waluyo, Zubairi Djoerban, Tommy Hariman Siddiq

**Affiliations:** 1Department of Internal Medicine, Faculty of Medicine, Universitas Indonesia—Cipto Mangunkusumo Hospital, Jakarta 10430, Indonesia; 2Allergy and Clinical Immunology Division, Department of Internal Medicine, Faculty of Medicine, Universitas Indonesia/Cipto Mangunkusumo Hospital, Jakarta 10430, Indonesia; 3Faculty of Psychology, Universitas Indonesia, Depok 16424, Indonesia; 4Department of Non-Oncology Internal Medicine, Dharmais National Cancer Hospital, Jakarta 11420, Indonesia; 5Kramat 128 Hospital, Jakarta 10430, Indonesia; 6Faculty of Psychology and Education, Universitas Al Azhar Indonesia, Jakarta 12110, Indonesia

**Keywords:** COVID-19 vaccine, integrated behavior model, intention, people living with HIV

## Abstract

People living with HIV (PLHIV) are considered a high-risk population for developing a severe form of COVID-19. Vaccination is still one of the most important modalities in combating the disease due to the lack of an effective treatment. This multicenter study was performed from September to December 2021 with the aim to analyze the intention of PLHIV to receive the COVID-19 vaccination based on an integrated behavior model (IBM) in Indonesia. Of a total of 470 participants, 75.6% of patients were intent to be vaccinated. The model that was designed in this study explains 43.4% of the variance in intention to be vaccinated against COVID-19 in PLHIV (adjusted R^2^ = 0.434). Furthermore, the determinants used included instrumental attitude (β = 0.127, *p* < 0.05), subjective norm (β = 0.497, *p* < 0.01), and perceived behavioral control (β = 0.116, *p* < 0.01). This study concluded that an IBM could predict the intention of PLHIV to receive COVID-19 vaccination.

## 1. Introduction

The coronavirus disease of 2019 (COVID-19) is an infectious disease caused by severe acute respiratory syndrome coronavirus 2 (SARS-CoV−2). This disease remains a global public health problem that has affected over 610 million people and resulted in over 6.5 million deaths globally [1]. From the beginning of pandemic to November 2022, 145 thousand deaths due to COVID-19 were reported in Indonesia [2]. Despite most cases of COVID-19 being clinically mild or asymptomatic, older patients and those with multiple comorbidities have an increased risk of severe disease and death [3,4].

Efforts to minimize the incidence and severity of COVID-19 are mainly through vaccination due to the lack of an effective curative treatment [5,6]. Despite the remarkable production of vaccines and unprecedented heights reached in research and manufacturing speed, the worldwide threat of this virus is far from ending, especially for high-risk groups. According to the most recent Global COVID Access Tracker data, approximately one-quarter of the world’s most vulnerable populations still require a primary vaccination series against COVID-19 [7].

People with comorbidities are classified as vulnerable, while people living with HIV (PLHIV) are considered to be at high risk of developing COVID-19 infection due to their impaired immunity. PLHIV with immune suppression, unsuppressed HIV RNA viral load, or comorbidities are at increased risk of severe disease and death from COVID-19 infection [4,8]. A recent meta-analysis reported that PLHIV have 1.3- and 2.3-fold increased risks of severe disease and death, respectively, compared to patients without HIV infection [4]. Another meta-analysis also indicated that HIV is associated with a 1.8-fold increased risk of death due to COVID-19 infection [9].

Research on COVID-19 vaccine acceptance and intention among PLHIV has been conducted in several countries. The results of these studies have shown that there is still apprehension and hesitancy in this population [10,11,12]. One of the many reasons behind the reluctance of PLHIV has been restricted access to vaccination [13]. At the beginning of the COVID-19 vaccination program in Indonesia, the required pre-vaccine evaluation form contained a question about HIV status [14,15]. This question may have induced hesitation among PLHIV to receive vaccination. Therefore, this study recognizes that it is essential to further examine the beliefs of PLHIV about COVID-19 vaccination and develop strategies to encourage its uptake [16].

The use of theoretical behavior models can be valuable in predicting intention to be vaccinated and for understanding the determinants of acceptance or hesitancy [17]. The integrated behavior model (IBM) is a framework for understanding and influencing human behavior centered on intention [18]. Theoretical behavior models have been widely used to determine the intention of COVID-19 vaccination in the general population, medical and nursing students, and healthcare workers [5,17,19,20]. 

By analyzing the behavior of a population, it should be possible to determine the most likely significant IBM components that play a role in intention to receive a COVID-19 vaccination. However, there are still limited studies on the intention of PLHIV, and none of these studies used an IBM approach. Therefore, this study aimed to analyze the attitudes and beliefs of PLHIV with regards to COVID-19 vaccination and identify the determinants influencing the decision to be vaccinated using an IBM framework.

## 2. Materials and Methods

### 2.1. Theoretical Framework

Montaño and colleagues developed the IBM by combining the theory of reasoned action (TRA), the theory of planned behavior (TPB), and other notable behavioral theories. The IBM incorporates the TRA, TPB, the health belief model (HBM), and social cognitive theory [21]. The model’s fundamental construct is behavior, which is defined as a single observable action performed by an individual. The intention to conduct the behavior, as in the TRA/TPB, is the essential driver of behavior. Thus, there should be a high degree of consistency between intention and behavior. According to the paradigm, three theoretical factors (attitude, perceived norm, and personal agency) primarily influence behavioral intention [21,22,23].

### 2.2. Study Design

This was a cross-sectional study conducted from September to December 2021 using a paper-based survey and an online survey. The subjects were PLHIV of at least 18 years of age who had consented to the questionnaires and were able to complete them. The participants were recruited from the outpatient clinics of three HIV centers in Jakarta: Cipto Mangunkusumo Hospital, Dharmais National Cancer Hospital, and Kramat 128 Hospital.

The study was conducted in two phases, (1) a qualitative research-elicitation phase to identify COVID-19 vaccination concerns among a representative sample of PLHIV in the study sites, and (2) a cross-sectional quantitative survey. First, individual qualitative interviews were conducted with nine PLHIV in the Indonesian language with questions structured around IBM components. The subjects were asked to consider receiving COVID-19 vaccination, and then report their sentiments and views regarding the outcomes and sources of normative influence, as well as the challenges and facilitators of vaccination. Consent analysis of the transcribed interviews presented their sentiments, sources of normative influence towards vaccination, obstacles, and facilitators of vaccination. Finally, a content analysis was performed for ten instrumental attitudes, five experiential attitudes, six subjective norms, three self-efficacies, and three perceived behavioral control statements. 

### 2.3. Survey Instrument

Each item of the IBM instrument was assessed using a 7-point bipolar scale. Behavioral intentions were measured using three items related to PLHIV intention to receive COVID-19 vaccination. These included “I hope to be able to be vaccinated against COVID-19 when the schedule is available”, “I want to be vaccinated against COVID-19 with the currently available vaccine”, and “I will continue to be vaccinated against COVID-19 even though there are obstacles to doing so.” Participants responded using a scale of strongly disagree (1) to strongly agree (7). The final result was the mean score of the three items. The mean score was then used to define intention. Values above 5 on this scale of 1 to 7 were classified as “intent to receive vaccination,” a value of 4 was classified as “undecided or neutral,” and below 3 was classified “not intent to receive vaccination.”

The ten instrumental and five experiential attitude statements were multiplied by a comparable item evaluating each sentence to measure attitude toward COVID-19 vaccination. The multiplicative scores for all items were added together. Six subjective norm statements were multiplied by a motivation score to comply with each statement to determine the subjective norm. Self-efficacy and perceived behavioral control were measured by multiplying each belief by a perceived power item. Since injunctive norms were not presented during the elicitation phase, this construct was not investigated.

In addition, data on age, gender, marital status, education status, occupation, antiretroviral therapy (ART) consumption, recent absolute cluster differentiation 4 (CD4) count, HIV risk factors, history of COVID-19 infection, first COVID-19 vaccination status, and history of vaccination other than COVID-19 were collected.

### 2.4. Recruitment

The eligible patients from the outpatient clinics were selected using simple random sampling, which was automatically generated using random.org. The chosen participants then received information about the study objectives and procedure and completed an informed consent, either written or electronic according to their preference. If a selected patient refused to give consent or had been recruited in the previous visit, we shifted recruitment to the next selected patient in line. 

The consent form included a disclaimer stating that participation was voluntary and refusal to participate had no implication. This study was conducted in accordance with the Declaration of Helsinki and approved by the Ethics Committee of the Faculty of Medicine, Universitas Indonesia.

### 2.5. Data Analysis

Data were analyzed using SPSS® Statistics 25 for Windows software (IBM Corp., Armonk, New York, USA). Descriptive analyses were conducted on background or sociodemographic factors and HIV characteristics. The calculated mean and standard deviation are presented for continuous variables, and categorical variables are presented as numbers and percentages. Syntax was used for common data transformation [24] to account for non-normality within the linear model framework and ensure statistical conclusion validity. The results of the evaluation of assumptions led to the transformation of the variables to reduce skewness and the number of outliers, as well as improve the normality, linearity, and homoscedasticity of residuals.

Analyses were conducted to identify the specific beliefs underlying the IBM constructs that best explained COVID-19 vaccination intention. The enter method was used to run hierarchical multiple regression on each IBM component linked with intention. Finally, statistical significance was evaluated at the *p* < 0.05 level. The correlation between level of intention and status of the first COVID-19 vaccination was also analyzed using the Kendall Tau-B method.

## 3. Results

### 3.1. Background Characteristics

A total of 4538 PLHIV were recruited during their clinic visits. After random sampling, 775 were invited to join the study. Due to time constraints, 295 PLHIV were unwilling to participate. The number of participants who completed the questionnaires at each study site were 204, 211, and 65 from Cipto Mangunkusumo Hospital, Dharmais National Cancer Hospital, and Kramat 128 Hospital, respectively. A total of 456 out of 480 participants primarily opted for paper-based questionnaires; however, 10 of these participants submitted incomplete IBM questionnaire data. As a result, 470 participants were included in the analysis, as shown in Figure 1.

Most participants were men (76.2%) with a median age of 41 years old (IQR = 19, range 18−76 years old). Seventy percent of participants had recent CD4 counts of more than 200 cells/mm^3^. The majority of participants were on antiretroviral therapy (96.4%), with a median duration of nine years (IQR = 7). Eighty-three participants (17.7%) had previously contracted COVID-19. The participants’ demographic, socioeconomic, and other clinical characteristics are summarized in Table 1.

### 3.2. Intention

The intention of PLHIV to be vaccinated against COVID-19 was found to be quite satisfactory, as 75.6% of participants had intention to receive vaccination, 20.6% were unsure or neutral, and 3.8% had no intention to receive vaccination. Cross-tabulation between intention to receive COVID-19 and the actual behavior of receiving the first COVID-19 vaccination exhibited a moderate association (Kendall’s tau-b = 0.226, *p* < 0.01). This indicates that there is a link between the intention of PLHIV to receive COVID-19 vaccination and their actual behavior in receiving it. 

### 3.3. IBM Constructs

Based on the internal consistency analysis and correlation with the cut-off value of 0.30, items that construct instrumental attitude, experiential attitude, subjective norm, self-efficacy, and perceived behavioral control exhibited positive correlation results with Cronbach’s alpha with values of 0.910, 0.768, 0.918, 0.652, and 0.714, respectively. Two items in instrumental attitude, one in experiential attitude, and one in perceived behavioral control were eliminated. Table 2 demonstrates the internal consistency and correlation between each item and the independent variable.

### 3.4. Final Regression Model

The final regression model included all constructs of the IBM. Data on subjective norms and perceived behavioral control were normally distributed, but data on instrumental attitude, experiential attitude, self-efficacy, and intention were not. There were no cases of missing data.

Table 3 displays the correlation between the variables, the unstandardized (B) and standardized regression coefficients (β), R, and R^2^ for the model analyzing COVID-19 vaccination intention. In the model, IBM items and intention were analyzed. It explained 43.4% of the variance in the intention of PLHIV to receive COVID-19 vaccination (adjusted R^2^ = 0.434). R for regression was significantly different from zero, F(5, 470) = 69.437, *p* < 0.001. 

The significant predictors of intention to receive COVID-19 vaccination in the model were instrumental attitude (*p* < 0.05), subjective norm (*p* < 0.01), and perceived behavior control (*p* < 0.01). In the model, these constructs were considered positive predictors of intention.

In instrumental attitude, the phrase “Getting the COVID-19 vaccination will reduce the severity of SARS-CoV−2 infection once I get infected.” had a significant effect (β = 0.199, *p* = 0.001). Further analysis of the subjective norms indicated that government program (β = 0.15, *p* = 0.002), doctors (β = 0.81, *p* = 0.011), religious leaders (β = 0.15, *p* = 0.012), and family (β = 0.202, *p* = 0.001) were all considerable factors in the intention of receiving COVID-19 vaccination. The results of multiple linear regression revealed that both items of the perceived behavioral control construct were significant items in the model: “How often have you been in a situation that made it easy for you to achieve what you wanted?” (β = 0.11, p = 0.036) and “How often does the belief ‘I can do it’ dominate your daily life?” (β = 0.36, *p* < 0.01). Further details of the IBM components are attached in Appendix A. 

## 4. Discussion

This is the first Indonesian study on PLHIV intention to receive vaccination against COVID-19. This multicenter study used a simple random sampling method to select representatives of the population and participants were allowed to respond using an online or an offline method. The population in this study was made up of relatively technology-illiterate people, who are underrepresented in many surveys. Most of the study participants (95%) preferred using paper-based questionnaires. In addition, this is the first study to use the IBM approach in PLHIV, which is the latest method of predicting intention to receive COVID-19 vaccination. However, 38% of patients initially selected refused to participate, primarily due to a lack of time during clinic visits. Many PLHIV only spend a short amount of time in the clinic for their routine ART visits. 

Most of the participants (75.6%) had the intention to receive COVID-19 vaccination. This finding is higher than a study conducted in 2021 in eight major cities in China, where only 57.2% of patients were willing to be vaccinated [10]. This number is also higher than the report from a study in the Middle East and North Africa between March and August 2021, which stated a vaccine acceptance rate of 64.6% among PLHIV [25]. The current study started in September 2021, several months after the initiation of the COVID-19 vaccination program for the general population in Indonesia. Therefore, hesitancy regarding its use in the PLHIV population and misinformation about the vaccines were still high. The vaccine recommendation in Indonesia initially comprised a question regarding HIV status in its screening form, including the respondent’s latest CD4 level [14,15]. This may have made many PLHIV uncertain about receiving the vaccination, not only because they would need to disclose their HIV status in the vaccine center, but also because it may have made them question their eligibility to get the vaccination.

Previous studies have reported that vaccination intention was lower in PLHIV compared to the non-HIV population [12]. To the authors’ knowledge, there is no study in Indonesia comparing the COVID-19 vaccination intention of PLHIV to that of non-PLHIV. However, our previous study found that PLHIV do have an understanding that they are more susceptible to COVID-19 infection. [26] A recent study reported that mortality and severity of COVID-19 were higher in PLHIV with lower CD4 counts [27]. In the current study, most patients had high CD4 counts. Nevertheless, the participants appeared to have a high sense of perceived risk of severe COVID-19 infection. This was shown by the significant level of acceptance that receiving the vaccination would reduce the severity of COVID-19 infection.

In our final model, instrumental attitude, perceived behavioral control, and subjective norms were the three constructs of IBM that predicted 43% PLHIV intention to receive the COVID-19 vaccination. Instrumental attitude is defined as a person’s overall favorableness or unfavorableness toward performing a behavior based on the cognitive dimension [21]. “Getting the COVID-19 vaccination will reduce the severity of SARS-CoV−2 infection once I get infected” was the only statement in the instrumental attitude construct that affected intention (β 0.645, *p* < 0.01). This finding correlates to a study in Italy, which found that PLHIV believed that their condition was associated with a higher risk of developing complications of COVID-19 infection (*p* = 0.013) [28]. A previous study in Indonesia using the health belief model on the general population also demonstrated a similar association between vaccination willingness and the perceived benefits of COVID-19 vaccination [29]. 

Ajzen et al. [21,30] added perceived behavioral control in the TPB model. This was performed to account for the reasoning that a person’s perception of control over behavioral performance, together with intention, is expected to directly affect behavior, mainly when volitional control is not high. Among PLHIV, volitional control over the choice of COVID-19 vaccination indicated by the PBC items appeared to be a significant predictor in this current study. The earliest COVID-19 vaccination recommendation released by the Ministry of Health and The Indonesian Society of Internal Medicine back in 2020 suggested against vaccinating PLHIV with a low CD4 count. It is possible that this led to PLHIV experiencing difficulties receiving a vaccination as vaccination centers likely suspended vaccines for PLHIV with unknown recent CD4 counts. This recommendation was cancelled in December 2020; however, it appears that it still causes confusion and apprehension for many healthcare workers [15].

It is interesting to note that the participants to this study still considered their loved ones’ advice when making the decision to receive a COVID-19 vaccination. This subjective norm was the most significant construct in the IBM framework related to vaccination intention (β = 0.497, Appendix A). The most important item of the subjective norms was found to be “My family expects me to receive the COVID-19 vaccination”. Government initiatives, advice from religious leaders, and suggestions from their doctors followed in the three next highest positions. This outcome shows very clearly how important it is to educate the patients and their families. The importance of clear and transparent information, particularly from the relevant authorities, is emphasized in this study.

This study has some limitations. First, about one-third of the selected PLHIV refused to participate in the study. Some of the reasons for this non-response error in survey were a lack of time, a lack of motivation, and fear of being registered as a HIV positive person. In terms of time, in all three centers, most PLHIV only plan to spend a short amount of time at the clinic for their routine ART visit. In addition, unvaccinated PLHIV might have a negative attitude towards vaccination, making them more likely to refuse to participate in the study. The study found a moderate correlation between intention to receive a vaccination and the actual action of getting the first COVID-19 vaccination. We could not find any study that indicated a direct association between intention and the actual behavior in receiving vaccination. Jaiswal et al., in fact, demonstrated a negative association between the COVID-19 vaccine hesitancy score and the intention to be vaccinated. COVID-19 vaccine hesitancy score was also negatively associated with COVID-19 vaccine uptake in PLHIV. COVID-19 vaccine hesitancy score also had a negative association with the intention to be vaccinated [31]. 

A second limitation is that our study was performed during the initial stages of the COVID-19 vaccination in the general population. Therefore, the questionnaire only asked about the first COVID-19 vaccination uptake. We do not have data regarding their actual behavior toward receiving the second or third COVID-19 vaccination. It is important to continuously analyze intention and actual vaccine uptake among PLHIV in line with developments in the pandemic situation. This is because uptake is highly correlated with vaccine safety, benefits, and real-life circumstances of the pandemic. However, we can use this IBM model to create a more focused approach to encourage the next COVID-19 vaccination or other new vaccines in the future.

The results of this current study could help policymakers and related institutions to form better recommendations and policies for COVID-19 vaccination in PLHIV. First and foremost, there needs to be focus on the benefit of the COVID-19 vaccination for PLHIV, particularly the reduction of the severity of SARS-CoV−2 infection. Second, PLHIV in this study had low perceived behavioral control; thus, they need external support to ensure a firm intention to receive COVID-19 vaccination. Aside from the government’s national program, endorsement from religious leaders and their HIV physicians is essential for this population. Involving family members in education regarding COVID-19 vaccination is also needed. In addition, providing PLHIV with relevant and evidence-based information would help strengthen their intention. 

## 5. Conclusions

This study has demonstrated the IBM model’s feasibility for the prediction of the intention of PLHIV to a receive COVID-19 vaccination. Among the components of the IBM, instrumental attitude, subjective norms, and perceived behavioral control were significantly correlated with vaccination intention. The most critical factor was the subjective norms. Therefore, the concerns of PLHIV should be put at ease by a profusion of relevant and evidence-based recommendations about vaccine benefits and safety.

## Figures and Tables

**Figure 1 vaccines-11-00296-f001:**
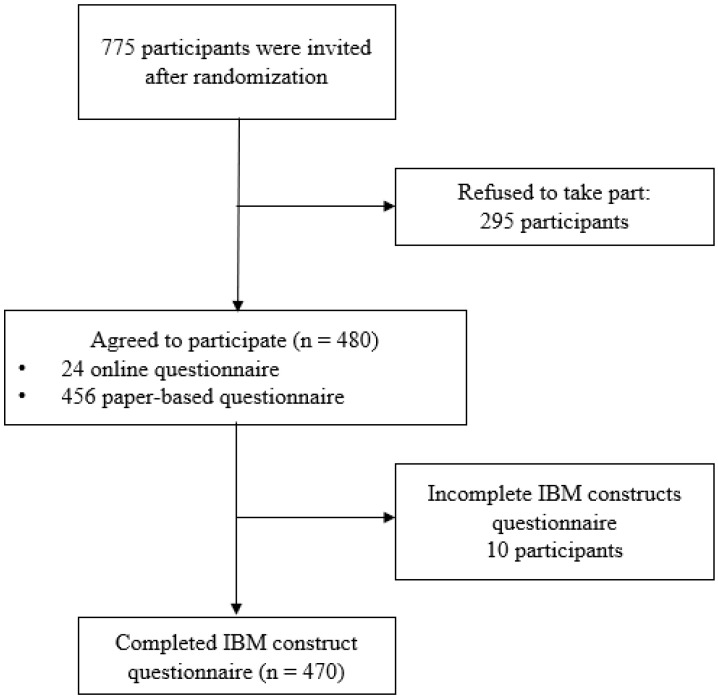
Flowchart of data collection.

**Table 1 vaccines-11-00296-t001:** Characteristics of study participants (*n* = 470).

Variables	Estimates, *n* (%)
Age in years, median (IQR)	41 (19)
Gender, *n* (%)	
Men	358 (76.2)
Women	112 (23.8)
Marital status, *n* (%)	
Married	230 (49.0)
Not married yet	174 (37)
Widow/widower	64 (13.6)
No data	2 (0.4)
Educational status, *n* (%)	
Low	46 (9.8)
Middle	195 (41.5)
High	226 (48.1)
No data	3 (0.6)
Occupation, *n* (%)	
Public servants	28 (6.00)
Private sectors	409 (87.0)
Not working	32 (6.8)
No data	1 (0.2)
ART consumption, *n* (%)	
On routine ART	453 (96.4) *
Not on ART	10 (2.1)
No data	7 (1.5)
Recent absolute CD4 count, *n* (%)	
<200 cells/mm^3^	54 (11.5)
≥200 cells/mm^3^	343 (72.9)
Unknown	73 (15.6)
HIV risk factors, *n* (%)	
Heterosexual	86 (18.3)
Homosexual	56 (11.9)
Intravenous drug user (IVDU)	114 (24.2)
Others	20 (4.3)
Not willing to inform	70 (14.9)
Unknown	124 (26.4)
History of COVID-19 infection, *n* (%)	83 (17.7)
Already recieved the first COVID-19 vaccine, *n* (%)	393 (83.6)
Ever received vaccine other than COVID-19, *n* (%) **	63 (13.4)

*: Median duration of ART consumption = 9 years (IQR 7); **: other than standard mandatory vaccinations during childhood.

**Table 2 vaccines-11-00296-t002:** IBM theoretical constructs and items related to the intention to receive COVID-19 vaccination.

Construct and Its Associated Items	Corrected-Item Total Correlation	Cronbach’s Alpha
Instrumental attitude		0.910
As long as I take ART regularly, I will be as healthy as a non-HIV individual.	0.607	
To end the pandemic, I will still follow the health protocols after receiving a COVID-19 vaccination.	0.821	
Getting the COVID-19 vaccination means I can protect my family.	0.780	
Getting the COVID-19 vaccination means I can protect myself.	0.761	
Vaccinating against COVID-19 is one of the efforts to end the pandemic.	0.835	
Getting the COVID-19 vaccination means I contribute to creating herd immunity.	0.832	
The COVID-19 vaccination can improve my immunity against COVID-19 infection.	0.527	
Getting the COVID-19 vaccination will reduce the severity of SARS-CoV−2 infection once I get infected.	0.645	
Experiential attitude		0.768
I feel more susceptible of getting COVID-19 infection compared to others.	0.456	
I am worried about receiving the COVID-19 vaccination because it might not be effective in preventing the infection.	0.588	
I am anxious of the unknown long-term side effects of COVID-19 vaccination.	0.598	
I am anxious of the side effects of COVID-19 vaccination.	0.619	
I am afraid of death if I get COVID-19 infection.	0.443	
Subjective norms		0.918
I receive the COVID-19 vaccine because it is a government program.	0.648	
My work colleagues expect me to receive the COVID-19 vaccination.	0.824	
My doctors expect me to receive the COVID-19 vaccination.	0.810	
My seniors at work expect me to receive the COVID-19 vaccination.	0.814	
The religious leaders that I respect expect me to receive the COVID-19 vaccination.	0.766	
My family expects me to receive the COVID-19 vaccination.	0.743	
Self-efficacy		0.652
In general, how often has the desire to be prioritized in gaining access to health services bothered you?	0.507	
How often do you feel that your situation is complicated by unclear administration and bureaucracy of health services?	0.511	
How much does it bother you if you do not pass the COVID-19 health screening?	0.378	
Perceived behavioral control		0.714
How often have you been in a situation that made it easy for you to achieve what you wanted?	0.555	
How often does the belief ‘*I can do it*’ dominate your daily life?	0.555	

**Table 3 vaccines-11-00296-t003:** Results of regression analysis between the intention to receive the COVID-19 vaccination with constructs of the IBM.

Variables	B	β	R	R^2^
Instrumental attitude	0.006 *	0.127	0.659	0.434
Experiential attitude	−0.062	−0.030		
Subjective norm	0.026 **	0.497		
Self-efficacy	−0.67	−0.030		
Perceived behavioral control	0.019 **	0.116		

*: *p* < 0.05; **: *p* < 0.01.

## Data Availability

All data in the manuscript can be obtained with reasonable request from the corresponding author.

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
