# Peer review of "Predicting Intention to Receive COVID-19 Vaccination in People Living with HIV using an Integrated Behavior Model"

_vaccines, 2023, doi:10.3390/vaccines11020296_

Round 1

Reviewer 1 Report

Overall a useful study that uses IBM constructs to identify factors involved in intention to receive covid 19 vaccine among PLHIV in Indonesia. The presentation of the methodology and results is generally clear, although there are some statements which could be further clarified.

My main concern is the apparent combination of participants who had received a previous covid 19 vaccine (and were presumably due for a second or subsequent dose), with those who had not previously received any covid vaccine. The current version of the paper is not explicit on the rationale for this combination, and does not distinguish between the two groups.

Some sections of the discussion were also somewhat confusing for the reviewer, particularly comparison with studies in Italy and Canada. Given the likelihood of significant differences in the populations of PLHIV and their medical care arrangements, the validity of these comparisons could be questioned. Other sections of the discussion focused on the PLHIV population in Indonesia are clearer.

More detailed comments follow

Abstract: Does not contain the sample size

Introduction

Line 65 ‘components’ – unclear what is meant here ? components of the IBM ?

Clear statement of purpose and rationale

Methods

Line 83 ’18 year old PLHIV’. Were all the subjects aged 18 years, or were they all older than 18 years ?

Line 113-115. The syntax here is a bit confusing, with different syntax for the two items that were multiplied by ‘perceived power’. It would be clearer to use similar syntax, or two state that two items were multiplied by perceived power – self-efficacy and perceived behavioural control.

Results

Table 1 includes one item of ‘already got covid 19 vaccine’. In this case, presumably the intention to be vaccinated refers to receiving a second or subsequent does of covid 19 vaccine. It would be clearer to include in the methods and the description of the study participants, that the majority had already received one dose of covid vaccine. It is possible that the motivation to receive a further dose of vaccine differs between individuals who had already received one dose, and those who were yet to receive any dose.

Line 164: 75.5% reported as ‘intended to be vaccinated’ – while Table 1 states that 83.6% had already received Covid 19 vaccine. It would be useful to compare the proportion previously vaccinated who intended to receive a further dose, with those not previously vaccinated.

Discussion

Line 208: ‘The population is reasonably representative of people struggling with technology’ Please provide some further explanation of what is meant by ‘people struggling with technology’ and why this population is representative of such people.

Line 248 The statement that ‘PLHIV have a more negative attitude towards Covid 19 vaccine’ does not seem consistent with the statement in line 246 that PLHIV believed that they were at higher risk of complications from covid 19. Given that there is no comparison with the attitudes of non PLHIV in this study, the comparison here referring to a Canadian study, is rather confusing.

Line 273 Limitations. The discussion should also reflect on the characteristics of the study population, and the implication of a population with a high proportion that had already received covid 19 vaccination

Line 280 In terms of recommendations, more focus on the implications of the importance of subjective norm, and consequently on the sources of information, as noted in lines 268 ff, rather than provision of information per se, would be useful in the recommendations.

Author Response

Response to Reviewer 1 Comments

Thank you for your keen observations and critical insights that have improved our manuscript. We hope that the manuscript in this revised state will be satisfactory. We look forward to more feedback or guidance as necessary. Please kindly see our responses below.

Comments and Suggestions for Authors

Overall a useful study that uses IBM constructs to identify factors involved in intention to receive covid 19 vaccine among PLHIV in Indonesia. The presentation of the methodology and results is generally clear, although there are some statements which could be further clarified.

Thank you for the appraisal. We are very appreciated the reviewer kindness in helping to improve the manuscript.

  1. My main concern is the apparent combination of participants who had received a previous covid 19 vaccine (and were presumably due for a second or subsequent dose), with those who had not previously received any covid vaccine. The current version of the paper is not explicit on the rationale for this combination, and does not distinguish between the two groups.

Response 1:

Sorry for the confusion. When we did a study, we asked whether the participant had got the 1st COVID-19 vaccine. We made a revision regarding the statement in table 1 from “already got COVID-19” vaccine to “already got the 1st COVID-19 vaccine”. This change was based on the consideration the condition during the study. In that time, Indonesia has just started the 1st COVID-19 vaccine and in preparation of 2nd COVID-19 vaccine. Thus, we hesitate that PLHIV had accessed for the 2nd COVID-19 vaccine.This study is initially meant to explore the intention and attitudes of PLHIV, regardless of their vaccination status or initial acceptance.

  1. Some sections of the discussion were also somewhat confusing for the reviewer, particularly comparison with studies in Italy and Canada. Given the likelihood of significant differences in the populations of PLHIV and their medical care arrangements, the validity of these comparisons could be questioned. Other sections of the discussion focused on the PLHIV population in Indonesia are clearer.

Response 2:

Thank you for the suggestion. We agreed to remove the dicsussion regarding studies in Italy and Canada in line 238.

More detailed comments follow

  1. Abstract: Does not contain the sample size

Reponse 3:

Thank you for the detail correction. We added the number in line 21.

 Introduction

  1. Line 65 ‘components’ – unclear what is meant here ? components of the IBM ?

Response 4:

Yes, that was correct, we refer to components of the IBM. We made a correction in line 65.

Methods

  1. Line 83 ’18 year old PLHIV’. Were all the subjects aged 18 years, or were they all older than 18 years ?

Response 5:

We apologize for the unclear statement. What we meant are they were 18 years old and older. We made a correction in line 83.

  1. Line 113-115. The syntax here is a bit confusing, with different syntax for the two items that were multiplied by ‘perceived power’. It would be clearer to use similar syntax, or two state that two items were multiplied by perceived power – self-efficacy and perceived behavioural control.

Response 6:

We made a revision in line 113.

Results

  1. Table 1 includes one item of ‘already got covid 19 vaccine’. In this case, presumably the intention to be vaccinated refers to receiving a second or subsequent does of covid 19 vaccine. It would be clearer to include in the methods and the description of the study participants, that the majority had already received one dose of covid vaccine. It is possible that the motivation to receive a further dose of vaccine differs between individuals who had already received one dose, and those who were yet to receive any dose.

Response 7:

We apologize for the confusion. We actually only ask whether they got the 1st COVID-19 vaccine. We added detailed description in line 117.

  1. Line 164: 75.5% reported as ‘intended to be vaccinated’ – while Table 1 states that 83.6% had already received Covid 19 vaccine. It would be useful to compare the proportion previously vaccinated who intended to receive a further dose, with those not previously vaccinated.

Reponse 8:

Thank you for the suggestion. We did correlation between intention and behavior covid-19 vaccination. There was a moderate correlation (value 0.22, p<0.01) between intention and already got 1st COVID-19 vaccination group. The description in line 172.

Discussion

  1. Line 208: ‘The population is reasonably representative of people struggling with technology’ Please provide some further explanation of what is meant by ‘people struggling with technology’ and why this population is representative of such people.

Response 9:

We sorry for the confusion. What we mean of ‘people struggling with technology’ is technology illiterate. In fact, the majority of those who were agree to participate (95%) preferred the use of paper-based questionnaires.

During questionnaire development we found out that many patients were struggling with online survey using their gadgets, alligately we assume their technology illiteracy. Therefore, we decided to use two type of survey, online and paper-based.

We made a correction in line 220.

  1. Line 248 The statement that ‘PLHIV have a more negative attitude towards Covid 19 vaccine’ does not seem consistent with the statement in line 246 that PLHIV believed that they were at higher risk of complications from covid 19. Given that there is no comparison with the attitudes of non PLHIV in this study, the comparison here referring to a Canadian study, is rather confusing.

Response 10:

We sorry for the confusion. We decided to take out the sentence.

  1. Line 273 Limitations. The discussion should also reflect on the characteristics of the study population, and the implication of a population with a high proportion that had already received covid 19 vaccination

Response 11:

Thank you for the suggestion.  We totally revised the paragraph about limitation in line 277 with the analysis of correlation between the intention and already got 1st COVID-19 vaccination.

  1. Line 280 In terms of recommendations, more focus on the implications of the importance of subjective norm, and consequently on the sources of information, as noted in lines 268 ff, rather than provision of information per se, would be useful in the recommendations.

Response 12:

Thank you for the suggestion. We added more focus recommendations in line 290.

Reviewer 2 Report

People living with HIV (PLHIV) are vulnerable to a variety of infectious and non-infectious diseases including COVID-19 because of immune system incompetence. Vaccination is absolutely essential for HIV-positive people. However, the willingness to vaccination among such people is often reduced.

The authors presented a study of COVID-19 vaccine acceptance and intention in 470 PLHIV from 3 clinics in Indonesia. The authors suggested that the results could be helpful for health policymakers and related institutions to create better recommendations and policies about COVID-19 infection in the PLHIV.

The manuscript presents the results of a sociological survey; it does not contain any experimental research or drug development. The study design is based on the Integrated Behavior Model (IBM), which incorporates the Theory of Reasoned Action, the Theory of Planned Behavior, Health Belief Model, and Social Cognitive Theory. The study design and data analysis are adequate.

The results showed that IBM can predict the intention of PLHIV to receive the COVID vaccine.

There are some questions about the data. The data on study participants are a bit controversial. So, line 151 and Fig.1 contain information about 470 subjects included in the analysis, while as stated earlier by the authors, “The number of participants who completed the questionnaire were 237, 211, and 65 participants from Cipto Mangunkusumo Hospital, Dharmais National Cancer Hospital, and Kramat 128 Hospital, respectively”, what in total is 513 people. This difference is not clear.

At the moment of study, the majority of participants (83.6%) had been already vaccinated against COVID-19, which may cause a bias.  According to the survey results most of the participants (75.5%) showed the intention of receiving a COVID-19 vaccine. Unvaccinated people may demonstrate different attitudes. Unvaccinated people or people who have a negative attitude towards vaccination could more likely refuse to participate in the study. I propose to discuss this in the article.

Summarizing, the manuscript can be accepted for publication after minor revision of the remarks mentioned above, while the question remains how does this topic correspond to the publication policy and the profile of the journal?

Author Response

Response to Reviewer 2 Comments

Thank you for your keen observations and critical insights that have improved our manuscript. We hope that the manuscript in this revised state will be satisfactory. We look forward to more feedback or guidance as necessary. Please kindly see our responses below.

Comments and Suggestions for Authors

People living with HIV (PLHIV) are vulnerable to a variety of infectious and non-infectious diseases including COVID-19 because of immune system incompetence. Vaccination is absolutely essential for HIV-positive people. However, the willingness to vaccination among such people is often reduced.

The authors presented a study of COVID-19 vaccine acceptance and intention in 470 PLHIV from 3 clinics in Indonesia. The authors suggested that the results could be helpful for health policymakers and related institutions to create better recommendations and policies about COVID-19 infection in the PLHIV.

The manuscript presents the results of a sociological survey; it does not contain any experimental research or drug development. The study design is based on the Integrated Behavior Model (IBM), which incorporates the Theory of Reasoned Action, the Theory of Planned Behavior, Health Belief Model, and Social Cognitive Theory. The study design and data analysis are adequate.

The results showed that IBM can predict the intention of PLHIV to receive the COVID vaccine.

Thank you for the appraisal. We are very appreciated the reviewer kindness in helping improve the manuscript. We hope that the manuscripst is now acceptable.

  1. There are some questions about the data. The data on study participants are a bit controversial. So, line 151 and Fig.1 contain information about 470 subjects included in the analysis, while as stated earlier by the authors, “The number of participants who completed the questionnaire were 237, 211, and 65 participants from Cipto Mangunkusumo Hospital, Dharmais National Cancer Hospital, and Kramat 128 Hospital, respectively”, what in total is 513 people. This difference is not clear.

Response 1:

Thank you for detail correction. We apologize for the discrepancy number. The correct number of participant in Ciptomangunkusumo Hospital was 204 instead of 237. We have corrected this number in line 153.

  1. At the moment of study, the majority of participants (83.6%) had been already vaccinated against COVID-19, which may cause a bias. According to the survey results most of the participants (75.5%) showed the intention of receiving a COVID-19 vaccine. Unvaccinated people may demonstrate different attitudes. Unvaccinated people or people who have a negative attitude towards vaccination could more likely refuse to participate in the study. I propose to discuss this in the article.

Response 2:

Thank you for the input. We made a revision regarding the statement in table 1 from “already got COVID-19” vaccine to “already got the 1st COVID-19 vaccine”. This change was based on the consideration the condition during the study. In that time, Indonesia has just started the 1st COVID-19 vaccine and in preparation of 2nd COVID-19 vaccine. Thus, we hesitate that PLHIV had accessed for the 2nd COVID-19 vaccine.

In addition, We did Kendall’s Tau-b to identify correlation between intention group and already got the 1st COVID-19 vaccine group. From the analysis, we conclude that there was a correlation between intention and 1st COVID-19 vaccination with value 0.226 (p<0.01). It means the group of intended to be vaccinated was truly did a vaccination and vice versa. We believe this model could predict PLHIV in term of do the second, the third or more COVID-19 vaccine and can be a benchmark for the other vaccine in the future. We added the description in line 166.

According to reviewer recommendation, we added a selection bias in our limitation in line 278.

Summarizing, the manuscript can be accepted for publication after minor revision of the remarks mentioned above, while the question remains how does this topic correspond to the publication policy and the profile of the journal?

Round 2

Reviewer 1 Report

Overall the issues raised in the review of the initial draft have been adequately addressed. In particular, the issue of receipt of a first vaccine dose has been clarified and better addressed; and more specific recommendations based on the results of the study have been provided, which strengthen the usefulness of the study.

More detailed comments follow, indicating where the comments on the first version have been addressed.

Abstract: Does not contain the sample size

-        addressed

Introduction

Line 65 ‘components’ – unclear what is meant here ? components of the IBM ?

-        addressed

Clear statement of purpose and rationale

Methods

Line 83 ’18 year old PLHIV’. Were all the subjects aged 18 years, or were they all older than 18 years ?

-        addressed; would have been useful to include the entire age range

Line 113-115. The syntax here is a bit confusing, with different syntax for the two items that were multiplied by ‘perceived power’. It would be clearer to use similar syntax, or two state that two items were multiplied by perceived power – self-efficacy and perceived behavioural control.

-        The syntax has been addressed

-        Further information has been provided including clarification on the extent of data on other relevant factors, such as previous covid 19 vaccination or infection.

Results

Table 1 includes one item of ‘already got covid 19 vaccine’. In this case, presumably the intention to be vaccinated refers to receiving a second or subsequent does of covid 19 vaccine. It would be clearer to include in the methods and the description of the study participants, that the majority had already received one dose of covid vaccine. It is possible that the motivation to receive a further dose of vaccine differs between individuals who had already received one dose, and those who were yet to receive any dose.

-        included in Table 1, clarifying that majority had received a first dose of covid vaccine

Line 164: 75.5% reported as ‘intended to be vaccinated’ – while Table 1 states that 83.6% had already received Covid 19 vaccine. It would be useful to compare the proportion previously vaccinated who intended to receive a further dose, with those not previously vaccinated.

-        Additional information regarding relationship with previous vaccination and intention to receive vaccine provided.

Discussion

Line 208: ‘The population is reasonably representative of people struggling with technology’ Please provide some further explanation of what is meant by ‘people struggling with technology’ and why this population is representative of such people.

-        Additional information and explanation provided

Line 248 The statement that ‘PLHIV have a more negative attitude towards Covid 19 vaccine’ does not seem consistent with the statement in line 246 that PLHIV believed that they were at higher risk of complications from covid 19. Given that there is no comparison with the attitudes of non PLHIV in this study, the comparison here referring to a Canadian study, is rather confusing.

-        Reference to a study in Italy deleted.

-        This statement has been deleted.

Line 273 Limitations. The discussion should also reflect on the characteristics of the study population, and the implication of a population with a high proportion that had already received covid 19 vaccination

-        This section has been revised, to include reference to the proportion who refused to participate (although there is no particular reason to conclude that this might have biased the results);and to include reference to the correlation between intention and receipt of first dose of vaccine.

-        This section also clarifies that receipt of a second dose is not known.

Line 280 In terms of recommendations, more focus on the implications of the importance of subjective norm, and consequently on the sources of information, as noted in lines 268 ff, rather than provision of information per se, would be useful in the recommendations.

-        Recommendations have also been revised with more specific and targeted behavioural interventions included.

Author Response

We are thankful for your astute observations and insightful feedback, which helped us to improve our manuscript.

Overall the issues raised in the review of the initial draft have been adequately addressed. In particular, the issue of receipt of a first vaccine dose has been clarified and better addressed; and more specific recommendations based on the results of the study have been provided, which strengthen the usefulness of the study. 

More detailed comments follow, indicating where the comments on the first version have been addressed.

Methods

Line 83 ’18 year old PLHIV’. Were all the subjects aged 18 years, or were they all older than 18 years ?

-        1. addressed; would have been useful to include the entire age range

Response 1

  • Thank you for the suggestion. We did not limit the highest age in inclusion criteria however we added participant range in the result (18-76 years old) line 159.

Discussion

Line 273 Limitations. The discussion should also reflect on the characteristics of the study population, and the implication of a population with a high proportion that had already received covid 19 vaccination

-        2. This section has been revised, to include reference to the proportion who refused to participate (although there is no particular reason to conclude that this might have biased the results);and to include reference to the correlation between intention and receipt of first dose of vaccine.

Response 2

  • Thank you for the suggestion. We decided to change the paragraph.
  • We consider this is as a unit non-response error caused by lack of time, not motivated and fear of being registered. As the consequences there could be difference in characteristics between the participant and non-participant, at least some of the items in the survey.

Reference:

Biemer PP and Lyberg LE. Introduction to survey quality. United States of America:John Willey&Sons;2003.

  • COVID-19 vaccine. However, Jaiswal, et al. demonstrated negative association between COVID-19 vac-cine hesitancy score with intention to get vaccine. COVID-19 vaccine hesitancy score also had negative association with COVID-19 vaccine uptake in PLHIV. COVID-19 vaccine hesitancy score also had negative association with intention to get vaccine. We added the paragraph in line 282.
